# Global Belief Recursive Neural Networks

**Romain Paulus, Richard Socher**[*]
MetaMind
Palo Alto, CA
{romain,richard}@metamind.io

**Christopher D. Manning**
Stanford University
353 Serra Mall
Stanford, CA 94305
manning@stanford.edu

## Abstract

Recursive Neural Networks have recently obtained state of the art performance on several natural language processing tasks. However, because of their feedforward architecture they cannot correctly predict phrase or word labels that are determined by context. This is a problem in tasks such as aspect-specific sentiment classification which tries to, for instance, predict that the word *Android* is positive in the sentence *Android beats iOS*. We introduce global belief recursive neural networks (GB-RNNs) which are based on the idea of extending purely feedforward neural networks to include one feedbackward step during inference. This allows phrase level predictions and representations to give feedback to words. We show the effectiveness of this model on the task of contextual sentiment analysis. We also show that dropout can improve RNN training and that a combination of unsupervised and supervised word vector representations performs better than either alone. The feedbackward step improves F1 performance by 3% over the standard RNN on this task, obtains state-of-the-art performance on the SemEval 2013 challenge and can accurately predict the sentiment of specific entities.

## 1  Introduction

Models of natural language need the ability to compose the meaning of words and phrases in order to understand complex utterances such as facts, multi-word entities, sentences or stories. There has recently been a lot of work extending single word semantic vector spaces [27, 11, 15] to compositional models of bigrams [16, 29] or phrases of arbitrary length [25, 28, 24, 10]. Work in this area so far has focused on computing the meaning of longer phrases in purely feedforward types of architectures in which the meaning of the shorter constituents that are being composed is not altered. However, a full treatment of semantic interpretation cannot be achieved without taking into consideration that the meaning of words and phrases can also change once the sentence context is observed. Take for instance the sentence in Fig. 1: *The Android's screen is better than the iPhone's*. All current recursive deep learning sentiment models [26] would attempt to classify the phrase *The Android's screen* or *than the iPhone's*, both of which are simply neutral. The sentiment of the overall sentence is undefined; it depends on which of the entities the user of the sentiment analysis cares about. Generally, for many analyses of social media text, users are indeed most interested in the sentiment directed towards a specific entity or phrase.

In order to solve the contextual classification problem in general and aspect-specific sentiment classification in particular, we introduce global belief recursive neural networks (GB-RNN). These models generalize purely feedforward recursive neural networks (RNNs) by including a feedbackward step at inference time. The backward computation uses the representations from both steps in its recursion and allows all phrases, to update their prediction based on the global context of the sentence. Unlike recurrent neural networks or window-based methods [5] the important context can be many

---

[*]Part of this research was performed while the author was at Stanford University.

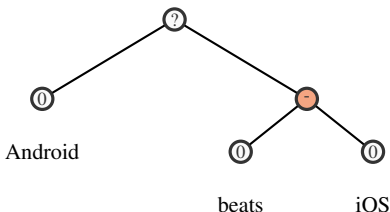

Figure 1: Illustration of the problem of sentiment classification that uses only the phrase to be labeled and ignores the context. The word *Android* is neutral in isolation but becomes positive in context.

words away from the phrase that is to be labeled. This will allow models to correctly classify that in the sentence of Fig. 1, *Android* is described with positive sentiment and *iOS* was not. Neither was possible to determine only from their respective phrases in isolation.

In order to validate the GB-RNN's ability to contextually disambiguate sentiment on real text, we use the Twitter dataset and annotations from Semeval Challenge 2013 Task 2.[1] The GB-RNN outperforms both the standard RNN and all other baselines, as well the winner of the Sentiment competition of SemEval 2013, showing that it can successfully make use of surrounding context.

## 2   Related Work

**Neural word vectors** One common way to represent words is to use distributional word vectors [27] learned via dimensionality reduction of large co-occurrence matrices over documents (as in latent semantic analysis [13]), local context windows [15, 18] or combinations of both [11]. Words with similar meanings are close to each other in the vector space. Since unsupervised word vectors computed from local context windows do not always encode task-specific information, such as sentiment, word vectors can also be fine-tuned to such specific tasks [5, 24]. We introduce a hybrid approach where some dimensions are obtained from an unsupervised model and others are learned for the supervised task. We show that this performs better than both the purely supervised and unsupervised semantic word vectors.

**Recursive Neural Networks** The idea of recursive neural networks (RNNs) for natural language processing (NLP) is to train a deep learning model that can be applied to inputs of any length. Unlike computer vision tasks, where it is easy to resize an image to a fixed number of pixels, natural sentences do not have a fixed size input. However, phrases and sentences have a grammatical structure that can be parsed as a binary tree [22].

Following this tree structure, we can assign a fixed-length vector to each word at the leaves of the tree, and combine word and phrase pairs recursively to create intermediate node vectors of the same length, eventually having one final vector representing the whole sentence [19, 25]. Multiple recursive combination functions have been explored, from linear transformation matrices to tensor products [26]. In this work, we use the simple single matrix RNN to combine node vectors at each recursive step.

**Bidirectional-recurrent and bidirectional-recursive neural networks.** Recurrent neural networks are a special case of recursive neural networks that operate on chains and not trees. Unlike recursive neural networks, they don't require a tree structure and are usually applied to time series. In a recurrent neural network, every node is combined with a summarized representation of the past nodes [8], and then the resulting combination will be forwarded to the next node. Bidirectional recurrent neural network architectures have also been explored [21] and usually compute representations independently from both ends of a time series.

Bidirectional recursive models [12, 14], developed in parallel with ours, extend the definition of the recursive neural netword by adding a backward propagation step, where information also flows from the tree root back to the leaves. We compare our model to theirs theoretically in the model section, and empirically in the experiments.

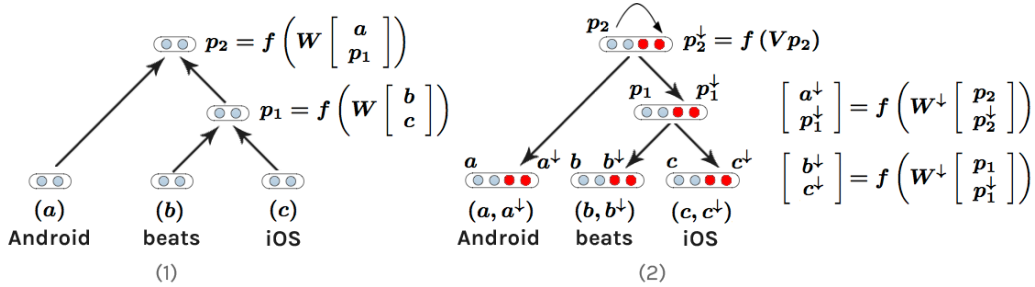

Figure 2: Propagation steps of the GB-RNN. Step 1 describes the standard RNN feedforward process, showing that the vector representation of "Android" is independent of the rest of the document. Step 2 computes additional vectors at each node (in red), using information from the higher level nodes in the tree (in blue), allowing "Android" and "iOS" to have different representations given the context.

[20] unfold the same autoencoder multiple times which gives it more representational power with the same number of parameters. Our model is different in that it takes into consideration more information at each step and can eventually make better local predictions by using global context.

**Sentiment analysis.** Sentiment analysis has been the subject of research for some time [4, 2, 3, 6, 1, 23]. Most approaches in sentiment analysis use "bag of words" representations that do not take the phrase structure into account but learn from word-level features. We explore our model's ability to determine contextual sentiment on Twitter, a social media platform.

## 3  Global Belief Recursive Neural Networks

In this section, we introduce a new model to compute context-dependent compositional vector representations of variable length phrases. These vectors are trained to be useful as features to classify each phrase and word. Fig. 2 shows an example phrase computation that we will describe in detail below. This section begins by motivating compositionality and context-dependence, followed by a definition of standard recursive neural networks. Next, we introduce our novel global belief model and hybrid unsupervised-supervised word vectors.

### 3.1  Context-Dependence as Motivation for Global Belief

A common simplifying assumption when mapping sentences into a feature vector is that word order does not matter ("bag of words"). However, this will prevent any detailed understanding of language as exemplified in Fig. 1, where the overall sentiment of the phrase "Android beats iOS", is unclear. Instead, we need an understanding of each phrase which leads us to deep recursive models.

The first step for mapping a sentence into a vector space is to parse them into a binary tree structure that captures the grammatical relationships between words. Such an input dependent binary tree then determines the architecture of a recursive neural network which will compute the hidden vectors in a bottom-up fashion starting with the word vectors. The resulting phrase vectors are given as features to a classifier. This standard RDL architecture works well for classifying the inherent or context-independent label of a phrase. For instance, it can correctly classify that *a not so beautiful day* is negative in sentiment. However, not all phrases have an inherent sentiment as shown in Fig. 1.

The GB-RNN addresses this issue by propagating information from the root node back to the leaf nodes as described below. There are other ways context can be incorporated such as with bi-directional recurrent neural networks or with window-based methods. Both of these methods, however, cannot incorporate information from words further away from the phrase to be labeled.

### 3.2  Standard Recursive Neural Networks

We first describe a simple recursive neural network that can be used for context-independent phrase-level classification. It can also be seen as the first step of a GB-RNN.

Assume, for now, that each word vector $a \in \mathbb{R}^n$ is obtained by sampling each element from a uniform distribution: $a_i \sim \mathcal{U}(-0.001, 0.001)$. All these vectors are columns of a large embedding matrix $L \in \mathbb{R}^{n \times |V|}$, where $|V|$ is the size of the vocabulary. All word vectors are learned together with the model.

For the example word vector sequence $(abc)$ of Fig. 2, the RNN equations become:

$$p_1 = f\left(W \begin{bmatrix} b \\ c \end{bmatrix}\right), \quad p_2 = f\left(W \begin{bmatrix} a \\ p_1 \end{bmatrix}\right), \tag{1}$$

where $W \in \mathbb{R}^{n \times 2n}$ is the matrix governing the composition and $f$ the non-linear activation function. Each node vector is the given as input to a $\mathrm{softmax}$ classifier for a classification task such as sentiment analysis.

### 3.3 GB-RNN: Global Belief Recursive Neural Networks

Our goal is to include contextual information in the recursive node vector representations. One simple solution would be to just include the $k$ context words to the left and right of each pair as in [25]. However, this will only work if the necessary context is at most $k$ words away. Furthermore, in order to capture more complex linguistic phenomena it may be necessary to allow for multiple words to compose the contextual shift in meaning. Instead, we will use the feedforward nodes from a standard RNN architecture and simply move back down the tree. This can also be interpreted as unfolding the tree and moving up its branches.

Hence, we keep the same Eq. 1 for computing the forward node vectors, but we introduce new feedbackward vectors, denoted with a down arrow $\downarrow$, at every level of the parse tree. Unlike the feedforward vectors, which were computed with a bottom-up recursive function, feedbackward vectors are computed with a top-down recursive function. The backwards pass starts at the root node and propagates all the way down to the single word vectors. At the root note, in our example the node $p_2$, we have:

$$p_2^{\downarrow} = f(V p_2), \tag{2}$$

where $V \in \mathbb{V}^{n_d \times n}$ so that all $\downarrow$-node vectors are $n_d$-dimensional. Starting from $p_2^{\downarrow}$, we recursively get $\downarrow$-node vectors for every node as we go down the tree:

$$\begin{bmatrix} a^{\downarrow} \\ p_1^{\downarrow} \end{bmatrix} = f\left(W^{\downarrow} \begin{bmatrix} p_2 \\ p_2^{\downarrow} \end{bmatrix}\right), \quad \begin{bmatrix} b^{\downarrow} \\ c^{\downarrow} \end{bmatrix} = f\left(W^{\downarrow} \begin{bmatrix} p_1 \\ p_1^{\downarrow} \end{bmatrix}\right) \tag{3}$$

where all $\downarrow$-vectors, are $n_d$-dimensional and hence $W^{\downarrow} \in \mathbb{R}^{(n+n_d) \times (n+n_d)}$ is a new de-composition matrix. Figure 2 step 2 illustrates this top-down recursive computation on our example. Once we have both feedforward and feedbackward vectors for a given node, we concatenate them and employ the standard $\mathrm{softmax}$ classifier to make the final prediction. For instance, the classification for word $a$ becomes: $y_a = \mathrm{softmax}\left(W_c \begin{bmatrix} a \\ a^{\downarrow} \end{bmatrix}\right)$, where we fold the bias into the $C$-class classifier weights $W_c \in \mathbb{R}^{C \times (n+1)}$.

At the root node, the equation for $x_{root}^{\downarrow}$ could be replaced by simply copying $x_{root}^{\downarrow} = x_{root}$. But there are two advantages of introducing a transform matrix $V$. First, it helps clearly differentiating features computed during the forward step and the backward step in multiplication with $W^{\downarrow}$. Second, it allows to use a different dimension for the $x^{\downarrow}$ vectors, which reduces the number of parameters in the $W^{\downarrow}$ and $W_{class}$ matrices, and adds more flexibility to the model. It also performs better empirically.

### 3.4 Hybrid Word Vector Representations

There are two ways to initialize the word vectors that are given as inputs to the RNN models. The simplest one is to initialize them to small random numbers as mentioned above and backpropagate error signals into them in order to have them capture the necessary information for the task at hand. This has the advantage of not requiring any other pre-training method and the vectors are sure to capture domain knowledge. However, the vectors are more likely to overfit and less likely to generalize well to words that have not been in the (usually smaller) labeled training set. Another approach

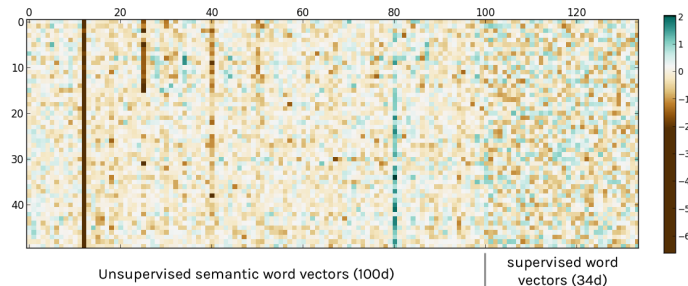

Figure 3: Hybrid unsupervised-supervised vector representations for the most frequent 50 words of the dataset. For each horizontal vector, the first 100 dimensions are trained on unlabeled twitter messages, and the last dimensions are trained on labeled contextual sentiment examples.

is to use unsupervised methods that learn semantic word vectors such as [18]. One then has the option to backpropagate task specific errors into these vectors or keep them at their initialization. Backpropagating into them still has the potential disadvantage of hurting generalization apart from slowing down training since it increases the number of parameters by a large amount (there are usually $100,000 \times 50$ many parameters in the embedding matrix $L$). Without propagating information however one has to hope that the unsupervised method really captures all the necessary semantic information which is often not the case for sentiment (which suffers from the antonym problem).

In this paper we propose to combine both ideas by representing each word as a concatenation of both unsupervised vectors that are kept at their initialization during training and adding a small additional vector into which we propagate the task specific error signal. This vector representation applies only to the feedforward word vectors and shold not be confused with the combination of the feedwordard and feedbackward node vectors in the $\mathrm{softmax}$.

Figure 3.4 shows the resulting word vectors trained on unlabeled documents on one part (the first 100 dimensions), and trained on labeled examples on the other part (the remaining dimensions).

## 3.5   Training

The GB-RNN is trained by using backpropagation through structure [9]. We train the parameters by optimizing the regularized cross-entropy error for labeled node vectors with mini-batched AdaGrad [7]. Since we don't have labels for every node of the training trees, we decided that unlabeled nodes do not add an additional error during training. For all models, we use a development set to cross-validate over regularization of the different weights, word vector size, mini-batch size, dropout probability and activation function (rectified linear or logistic function).

We also applied the dropout technique to improve training with high dimensional word vectors. Node vector units are randomly set to zero with a probability of 0.5 at each training step. Our experiments show that applying dropout in this way helps differentiating word vector units and hidden units, and leads to better performance. The high-dimensional hybrid word vectors that we introduced previously have obtained a higher accuracy than other word vectors with the use of dropout.

## 3.6   Comparison to Other Models

The idea of unfolding of neural networks is commonly used in autoencoders as well as in a recursive setting [23], in this setting the unfolding is only used during training and not at inference time to update the beliefs about the inputs.

Irsoy and Cardie [12] introduced a bidirectional RNN similar to ours. It employs the same standard feedforward RNN, but a different computation for the backward $\downarrow$ vectors. In practice, their model is defined by the same forward equations as ours. However, equation 3 which computes the backward vectors is instead:

$$\begin{bmatrix} b^{\downarrow} \\ c^{\downarrow} \end{bmatrix} = f\left( \begin{bmatrix} Vb + W_{lb}^{\downarrow}p_1^{\downarrow} \\ Vc + W_{rb}^{\downarrow}p_1^{\downarrow} \end{bmatrix} \right) \tag{4}$$

**Correct** FUSION's 5th General Meeting is tonight at 7 in ICS 213! **Come out** and carve pumpkins mid-quarter with us!

**Correct** I would **rather eat my left foot** then to be taking the SATs tomorrow

**Correct** Special THANKS to EVERYONE for coming out to Taboo Tuesday With DST tonight! It was **FUN&educational!!! :)** @XiEtaDST

**Correct** **Tough loss** for @statebaseball today. Good luck on Monday with selection Sunday

**Correct** I **got** the job at Claytons!(: I start Monday doing Sheetrock(: #MoneyMakin

**Correct** St Pattys is **no big deal** for me, no fucks are given, but Cinco De Mayo on the other hand .. thats my 2nd bday .

**Incorrect** "@Hannah_Sunder: The Walking Dead is just a great tv show" its **bad ass** just started to watch the 2nd season to catch up with the 3rd

Figure 4: Examples of predictions made by the GB-RNN for twitter documents. In this example, red phrases are negative and blue phrases are positive. On the last example, the model predicted incorrectly "bad ass" as negative.

Where $W_{lb}^{\downarrow}$ and $W_{rb}^{\downarrow}$ are two matrices with dimensions $n_d \times n_d$. For a better comparison with our model we rewrite Eq. 3 and make explicit the 4 blocks of $W^{\downarrow}$:

$$\text{Let} \quad W^{\downarrow} = \left[ \begin{array}{cc} W_{lf}^{\downarrow} & W_{lb}^{\downarrow} \\ W_{rf}^{\downarrow} & W_{rb}^{\downarrow} \end{array} \right] \quad , \text{then} \quad \left[ \begin{array}{c} b^{\downarrow} \\ c^{\downarrow} \end{array} \right] = f\left( \left[ \begin{array}{c} W_{lf}^{\downarrow}p_1 + W_{lb}^{\downarrow}p_1^{\downarrow} \\ W_{rf}^{\downarrow}p_1 + W_{rb}^{\downarrow}p_1^{\downarrow} \end{array} \right] \right), \quad (5)$$

where the dimensions of $W_{lf}^{\downarrow}$ and $W_{rf}^{\downarrow}$ are $n_d \times n$, and the dimensions of $W_{ld}^{\downarrow}$ and $W_{rd}^{\downarrow}$ are $n_d \times n_d$.

A closer comparison between Eqs. 4 and 5 reveals that both use a left and right forward transformation $W_{lf}^{\downarrow}p_1$ and $W_{rf}^{\downarrow}p_1$, but the other parts of the sums differ. In the bidirectional-RNN, the transformation of any children is defined by the forward parent and independent on its position (left or right node). Whereas our GB-RNN makes uses of both the forward and backward parent node. The intuition behind our choice is that using both nodes helps to push the model to disentangled the children from their backward parent vector. We also note that our model does not use the forward node vector for computing the backward node vector, but we find this not necessary since the $\mathrm{softmax}$ function already combines the two vectors.

Our model also has $n \cdot n_d$ more parameters to compute the feedbackward vectors than the bidirectional-RNN. The $W^{\downarrow}$ matrix of our model has $2n_d^2 + 2n \cdot n_d$ parameters, while the other model has a total of $2n_d^2 + n \cdot n_d$ parameters with the $W_{lf}^{\downarrow}$, $W_{rf}^{\downarrow}$ and $V$ matrices. We show in the next section that GB-RNN outperforms the bidirectional RNN in our experiments.

## 4 Experiments

We present a qualitative and quantitative analysis of the GB-RNN on a contextual sentiment classification task. The main dataset is provided by the SemEval 2013, Task 2 competition [17]. We outperform the winners of the 2013 challenge, as well as several baseline and model ablations.

### 4.1 Evaluation Dataset

The SemEval competition dataset is composed of tweets labeled for 3 different sentiment classes: positive, neutral and negative. The tweets in this dataset were split into a train (7862 labeled phrases), development (7862) and development-test (7862) set. The final test set is composed of 10681 examples. Fig. 4 shows example GB-RNN predictions on phrases marked for classification in this dataset. The development dataset consists only of tweets whereas the final evaluation dataset included also short text messages (SMS in the tables below).

Tweets were parsed using the Stanford Parser [22] which includes tokenizing of negations (e.g., *don't* becomes two tokens *do* and *n't*). We constrained the parser to keep each phrase labeled by the dataset inside its own subtree, so that each labeled example is represented by a single node and can be classified easily.

| Classifier | Feature Sets | Twitter 2013 (F1) | SMS 2013 (F1) |
|---|---|---|---|
| SVM | stemming, word cluster, SentiWordNet score, negation | 85.19 | 88.37 |
| SVM | POS, lexicon, negations, emoticons, elongated words, scores, syntactic dependency, PMI | 87.38 | 85.79 |
| SVM | punctuation, word $n$-grams, emoticons, character $n$-grams, elongated words, upper case, stopwords, phrase length, negation, phrase position, large sentiment lexicons, microblogging features | 88.93 | 88.00 |
| GB-RNN | parser, unsupervised word vectors (ensemble) | **89.41** | **88.40** |

Table 1: Comparison to the best Semeval 2013 Task 2 systems, their feature sets and F1 results on each dataset for predicting sentiment of phrases in context. The GB-RNN obtains state of the art performance on both datasets.

| Model | Twitter 2013 | SMS 2013 |
|---|---|---|
| Bigram Naive Bayes | 80.45 | 78.53 |
| Logistic Regression | 80.91 | 80.37 |
| SVM | 81.87 | 81.91 |
| RNN | 82.11 | 84.07 |
| Bidirectional-RNN (Irsoy and Cardie) | 85.77 | 84.77 |
| GB-RNN (best single model) | **86.80** | **87.15** |

Table 2: Comparison with baselines: F1 scores on the SemEval 2013 test datasets.

## 4.2 Comparison with Competition Systems

The first comparison is with several highly tuned systems from the SemEval 2013, Task 2 competition. The competition was scored by an average of positive and negative class F1 scores. Table 1 lists results for several methods, together with the resources and features used by each method. Most systems used a considerable amount of hand-crafted features. In contrast, the GB-RNN only needs a parser for the tree structure, unsupervised word vectors and training data. Since the competition allowed for external data we outline below the additional training data we use. Our best model is an ensemble of the top 5 GB-RNN models trained independently. Their predictions were then averaged to produce the final output.

## 4.3 Comparison with Baselines

Next we compare our single best model to several baselines and model ablations. We used the same hybrid word vectors with dropout training for the RNN, the bidirectional RNN and the GB-RNN. The best models were selected by cross-validating on the dev set for several hyper-parameters (word vectors dimension, hidden node vector dimension, number of training epochs, regularization parameters, activation function, training batch size and dropout probability) and we kept the models with the highest cross-validation accuracy. Table 2 shows these results. The most important comparison is against the purely feedforward RNN which does not take backward sentence context into account. This model performs over 5% worse than the GB-RNN.

For the logistic regression and Bigram Naive Bayes classification, each labeled phrase was taken as a separate example, removing the surrounding context. Another set of baselines used a context window for classification as well as the entire tweet as input to the classifier.

Optimal performance for the single best GB-RNN was achieved by using vector sizes of 130 dimensions (100 pre-trained, fixed word vectors and 30 trained on sentiment data), a mini-batch size of 30, dropout with $p = 0.5$ and sigmoid non-linearity. In table 3, we show that the concatenation of fixed, unsupervised vectors with additional randomly initialized, supervised vectors performs better than both methods.

## 4.4 Model Analysis: Additional Training Data

Because the competition allowed the usage of arbitrary resources we included as training data labeled unigrams and bigrams extracted from the NRC-Canada system's sentiment lexicon. Adding these additional training examples increased accuracy by 2%. Although this lexicon helps reduc-

| Word vectors | dimension | Twitter 2013 | SMS 2013 |
|---|---|---|---|
| supervised word vectors | 15 | 85.15 | 85.66 |
| semantic word vectors | 100 | 85.67 | 84.70 |
| hybrid word vectors | 100 + 34 | **86.80** | **87.15** |

Table 3: F1 score comparison of word vectors on the SemEval 2013 Task 2 test dataset.

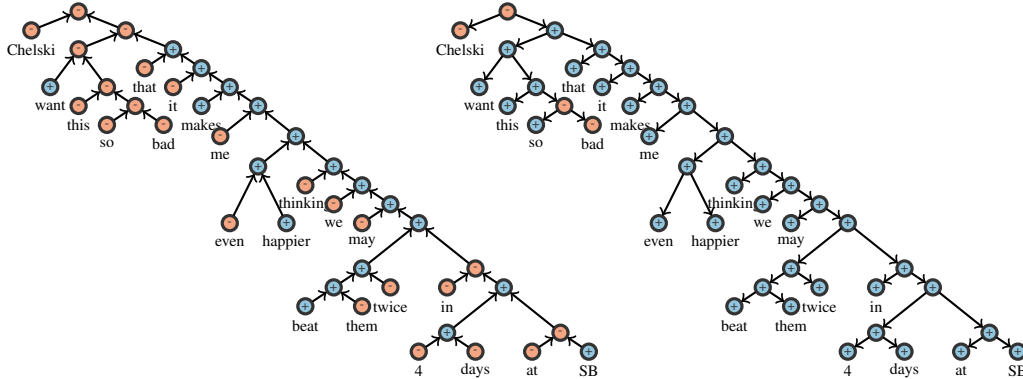

Figure 5: Change in sentiment predictions in the tweet *chelski want this so bad that it makes me even happier thinking we may beat them twice in 4 days at SB* between the RNN (left) and the GB-RNN (right). In particular, we can see the change for the phrase *want this so bad* where it is correctly predicted as positive with context.

ing the number of unknown tokens, it does not do a good job for training recursive composition functions, because each example is short.

We also included our own dataset composed 176,311 noisily labeled tweets (using heuristics such as smiley faces) as well as the movie reviews dataset from [26]. In both datasets the labels only denote the context-independent sentiment of a phrase or full sentence. Hence, we trained the final model in two steps: train the standard RNN, then train the full GB-RNN model on the smaller context-specific competition data. Training the GB-RNN jointly in this fashion gave a 1% accuracy improvement.

## 5 Conclusion

We introduced global belief recursive neural networks, applied to the task of contextual sentiment analysis. The idea of propagating beliefs through neural networks is a powerful and important piece for interpreting natural language. The applicability of this idea is more general than RNNs and can be helpful for a variety of NLP tasks such as word-sense disambiguation.

**Acknowledgments**

We thank the anonymous reviewers for their valuable comments.

## Footnotes

[1]http://www.cs.york.ac.uk/semeval-2013/task2/

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
