[Reviews · NeurIPS 2014]

Submitted by Assigned_Reviewer_2

This paper introduces "belief propagation recursive neural
networks". This kind of networks extends the "feedforward" (bottom-up)
recursive networks with a backward (top-down) step during
inference. This allows phrase level predictions and embeddings to
give feedback to word embeddings and labels.

This paper is overall well written. The model is well
motivated and described. This extends the idea of the
Bidirectional Recursive Neural Networks introduced by Irsoy
and Cardie 2008 (note that the reference to that paper is
incomplete). For instance one contribution is the introduction of hybrid word vectors.
Maybe this is the most important difference with previous work ?
Moreover, experimental comparisons are not completely fair: were the RNN and B-RNN also use hybrid vectors ? how the "best" models are tuned and selected ? How the single best is selected ? It is worth noticing that the others methods do not use an ensemble. These points must be clarified.

The term "belief propagation" is misleading and
maybe the term forward-backward could be well suited.

Section 3.5 (Training) is a little bit too short and could
provide more details. For instance, I guess that the training
use the back-propagation through structure algorithm (the paper
of Goller 1996 could be cited). This algorithm implies that
the recursive model is unfolded. In this case, this yields a
very deep network to "reach" the word embedding part that is
updated. If I understand correctly, I think that the authors
could provide more details on this very important step.

Section 3.6 is a little bit confused and the difference
between this model and the wok of Irsoy and Cardie must be
clarified.

Section 4.1 must be improved to be understood by readers
outside of the NLP community. In the task 2 of Semeval 2013
there are 2 subtasks: contextual and message polarity. I
guess that you address the contextual polarity task. With a
better presentation of the task, one can even understand the
parser constraints.

Maybe, sections 4.3 and 4.4 could be merged.

Summary: This paper is overall well written and describes an interesting variant of the Bidirectional recursive model.

Submitted by Assigned_Reviewer_3

In NLP, recursive neural networks (RNNs) have been used to produce representations snippets of text by recursively combining pairs of representations of words/shorter snippets. The process starts with the representation of words and proceeds up a pre-specified parse tree until the representation of the entire snippet is obtained at the root node. The unidirectional nature of this process, however, does not allow the information to be propagated down the tree from the larger to the smaller contexts. This paper introduces a bidirectional extension of RNNs in which the upward pass through the tree is followed by a downward pass, augmenting the representations of all the nodes with the information from larger contexts. The resulting system achieves state-of-the-art performance on Task 2 of SemEval 2012.

The extension of the established RNN-based approach proposed in the paper is simple, elegant and effective. Though the high-level idea is quite similar to the one from [18], the computation performed on the downward pass in the paper is different and the resulting performance appears to be superior.

The method is well motivated and clearly presented and the paper is nicely written in general. The experimental results appear to be excellent, but I have some concerns about the experimental setup. Was the result for the alternative bidirectional RNN model from [18] reported in Table 2 obtained using dropout and the additional training data? Were the hybrid vectors also used in that case? Also, while the ensemble result is impressive, it does not really belong in a table comparing the performance of individual models.

What does the "widely used mix of both" refer to in Section 4.4?
Summary: A well executed paper based on a simple and elegant idea.

Submitted by Assigned_Reviewer_29

This paper proposes using both the encoding and decoding phrase vectors of a recursive neural network auto-encoder in a sentiment classification task. The authors refer to this as belief propagation, although it is not an instance of this inference algorithm and as such this name serves to confuse the reader. In general this is a well written paper, however the motivation for the proposed model is vague and lacks a clear theoretical justification.

The key weakness of this paper is the evaluation. There is no reason to present an ensemble result in the context of this evaluation, none of the benchmarks are ensembles and this obfuscates the result. The exact composition of the models compared is also not clear. Key questions that should be clarified are: how were the models tuned and the ensemble/best models selected? How were the benchmarks in Table 2 tuned? Were they using the same augmented data as the BP-RNN? Were the RNN and Bidirectional-RNN also using hybrid word vectors?
In the absence of this information I would guess that the second entry in Table 3 is the most comparable to the benchmarks in Tables 1&2 when assessing the BP-RNN architecture.

Minor points:
- I am surprised that an off the shelf parser was used to parse tweets. I would be interested to know how accurate it is on this data.
- the Irsoy and Cardie citation is incomplete (as well as a number of others).
Summary: This paper presents a recursive neural network model for classifying phrasal sentiment in context. The results appear reasonable, but the model lacks a strong motivation and the experimental methodology is not entirely clear.
Author Feedback
Author rebuttal: We thank the reviewers for their reviews and time.

The comparison to other results in Table 2 is fair: We used the same hybrid word vectors with dropout training for the RNN, the bidirectional RNN and the BP-RNN. The best models were selected by cross-validating on the dev set for several hyper-parameters (word vectors dimension, hidden node vector dimension, number of training epochs, regularization parameters, activation function, training batch size and dropout probability) and we kept the models with the highest cross-validation accuracy.

The reason for mentioning the ensemble in our results is to compare with the Semeval 2013 results, which was a competition and included ensembles and combinations of models and a huge amount of manual feature engineering. We also include single model comparisons to related baselines.

We did not find any tweet parse trees dataset, so we were not able to train a new parser for tweets, nor could we evaluate the exact performance of the existing parser on twitter data. We assumed that the existing parsers along with the parser constraints would be sufficiently accurate for this task.

We will clarify the difference between our model and Irsoy & Cardie, and fix the incomplete citations and other minor problems.